# The effects of fresh embryo transfer and frozen-thawed embryo transfer on the perinatal outcomes of single fetuses from mothers with PCOS

Huizhen Li[1], Lihua Xu[1], Yanru Niu[2], Xia Zhu[1], Xiaomei Gao[1], Tianzhong Ma[1]*

1 Reproductive Medicine Center, Affiliated Hospital of Guangdong Medical University, Zhanjiang, Guangdong, China, 2 Laboratory of Bone Science, Affiliated Hospital of Guangdong Medical University, Zhanjiang, Guangdong, China

* tiann8283@163.com

**Data Availability Statement:** "All relevant data are within the paper and its Supporting Information files.

## Abstract

### Purpose

To investigate the effects of fresh embryo transfer and frozen-thawed embryo transfer on perinatal outcomes among patients with PCOS.

### Method

Patients who underwent in vitro fertilization and embryo transfer at the reproductive medicine center of the Affiliated Hospital of Guangdong Medical University from February 2013 to March 2021 were retrospectively analyzed. Patients were divided into the fresh embryo transfer group and frozen-thawed embryo transfer group according to whether fresh embryo transfer was performed. According to their conditions, patients were further classified into the ET-PCOS group (group A, n = 104), ET-non-PCOS group (group B, n = 212), FET-PCOS group (group C, n = 102), or FET-non-PCOS group (group D, n = 148); the general data, laboratory indicators and pregnancy outcomes of the patients were statistically analyzed, and the perinatal outcomes and related factors between the groups were compared and analyzed.

### Results

The level of $E_2$ on the HCG test day in the ET group was lower than that in the FET group. The natural birth rate of group D was lower than that of group A and group B, and the cesarean section rate was higher than that of group A and group B; the clinical pregnancy rate of group A was higher than that of group B and group D, and the difference was statistically significant (P < 0.05). There was no significant difference in the total abortion rate, early abortion rate or late abortion rate between the groups (P > 0.05). There was no significant difference in gestational age, neonatal sex or neonatal weight between the groups (P > 0.05). The incidence of placenta previa in Group B was significantly lower than that in Group D, and the difference was significant (P < 0.05). The incidence of fetal distress in Group B

**Funding:** This study was supported by funding from the National Natural Science Foundation of China (81300484) and the Natural Science Foundation of Guangdong Province, China (2022A1515010849). The funders is responsible for the publication costs of this manuscript. The funders had no role in study design, data collection and analysis, decision to publish, or preparation of the manuscript.

**Competing interests:** The authors have declared that no competing interests exist.

was significantly lower than that in Groups C and D, and the incidence of neonatal jaundice in Group D was significantly higher than that in Groups A and B (P < 0.05). In the multivariate analysis, the number of high-quality embryos was independent factors affecting clinical pregnancy, and the embryo transfer method was an independent factor affecting fetal distress and neonatal jaundice.

## Conclusion

Young PCOS patients without risk of OHSS have a high clinical pregnancy rate with fresh transplant cycles. PCOS disease itself has no significant effect on the perinatal outcomes of the mother or singleton infant. Frozen-thawed embryo transfer may increase the incidence of low placenta, fetal distress and neonatal jaundice.

## Introduction

Polycystic ovary syndrome (PCOS) is the most common endocrine disease in women of child-bearing age. PCOS is also a common cause of anovulatory infertility, accounting for approximately 80% of anovulatory infertility cases [1, 2]. With the increasing number of infertile people, assisted reproductive technology represents a new treatment for people with infertility [3]. The most critical difference between fresh embryo transfer and frozen-thawed embryo transfer (FET) is the continuous change in the hormone environment after COH. Compared with women with normal ovulation, PCOS patients have a stronger ovarian response. The supraphysiological level of estradiol caused by ovarian stimulation can affect embryo implantation, placental development and blood supply, resulting in adverse outcomes [4–7]. However, FET technology can reduce the supraphysiological state after ovarian stimulation and provide a more favorable intrauterine environment for embryo implantation and placenta formation [8]. However, it is still unclear whether there are differences in perinatal outcomes between women who undergo fresh and frozen-thawed embryo transfer. Therefore, the purpose of this study was to explore and analyze the effects of fresh embryo transfer and frozen-thawed embryo transfer on the perinatal outcomes of singletons from mothers with PCOS.

## Materials and methods

### Study design and participants

This was a retrospective analysis of patients who underwent IVF-ET-assisted conception at the Reproductive Center of the Affiliated Hospital of Guangdong Medical University between February 1, 2013, and March 27, 2021. All procedures performed in studies involving human participants were in accordance with the ethical standards of the institutional and national research committees and with the 1964 Helsinki declaration and its later amendments. This study was approved by the Ethics Committee of the Affiliated Hospital of Guangdong Medical University (PJKT2022-093). The study was initiated on 9/15/2022. Written informed consent was obtained from all patients, and all participants were informed that they could withdraw at any point during the study. The inclusion criteria for the PCOS group were as follows: (i) PCOS in accordance with the 2003 Rotterdam criteria; (ii) less than 35 years of age; (iii) retrieval of more than 5 eggs; (iv) selection of a long protocol; and (v) complete inspection index data in this center. The inclusion criteria for the control group were as follows: (i) less than 35 years of age; (ii) retrieval of more than 5 eggs; (iii) selection of a long protocol; (iv)

infertility caused by tubal factors; and (v) complete inspection index data in this center. Patients were excluded from the study if they met any of the following criteria: (i) reproductive system abnormalities or chromosome abnormalities; (ii) twin or multiple pregnancy; (iii) severe mental, endocrine or systemic disease; (iv) myoma of the uterus, endometriosis or tubal factor disease; (v) premature ovarian failure or decreased ovarian function; (vi) a history of uterine surgery; or (vii) missing data or missing visits for external reasons.

A total of 566 patients who met the inclusion criteria were included. Patients were divided into a fresh embryo transfer group and a frozen-thawed embryo transfer group according to whether fresh embryo transfer was performed. They were then divided into fresh embryo transfer and frozen-thawed embryo transfer groups. The patients were further classified into the ET-PCOS group (group A, n = 104), ET-non-PCOS group (group B, n = 212), FET-PCOS group (group C, n = 102), or FET-non-PCOS group (group D, n = 148).

## Treatment protocol

A long protocol for ovulation induction was used in the fresh cycle group. After the indicated standard was reached, 3.75 mg of long-acting GnRH-a (afolin, Huiling, Germany) was given on the 2nd—4th day of the menstrual cycle or mid-luteal phase, 0.8–1.875 mg GnRH-a was subcutaneously injected after ovulation on D7, or 0.05–0.1 mg GnRH-a was subcutaneously injected every day after ovulation on D7. When the pituitary downregulation standard [LH< 5 IU/L, $E_2$< 50 pg/mL, endometrial thickness < 5 mm] was used, exogenous gonadotropin 75–225 U/d [Gonaven, Merck, Germany; Prikon, Mushadong, USA; or Lishenbao, Zhuhai Lizhu] was given for ovulation induction. When the diameter of at least one follicle was 19 mm, the diameter of two follicles was 18 mm, or the diameter of three follicles was 17 mm, the blood $E_2$ level reached 250–300 pg/mL for each dominant follicle ($\geq$ 16 mm), or more than 60% for follicles greater than 16 mm, the injection of human chorionic gonadotropin (hCG) was 5000–10,000 IU that night. Fresh cleavage-stage embryos or blastocysts were transplanted on day 3 or 5 after fertilization.

In the frozen-thawed cycle group, 2~3 thawed good-quality embryos were selected for resuscitation and transplantation. The following three transplantation protocols were applied. (1) The natural cycle is suitable for patients with normal ovulation and an endometrial thickness > 8 mm. The follicles and endometrium were monitored on the 10th day of menstruation, and follicles >16 mm in size were monitored daily until ovulation. After ovulation, 40 mg progesterone was given daily to promote endometrial progression to the secretory phase, and the embryo was thawed and transferred 3 days later. (2) Hormone replacement cycles are suitable for patients with irregular menstruation, ovulation disorders or previous monitoring of endometrial thinning. From the second to the third day of menstruation, 6–8 mg of estradiol valerate was orally administered daily, and the dosage was adjusted according to the patient's condition. Endometrial thickness was monitored by B ultrasound after 10–12 days of treatment. If the endometrial thickness was < 8 mm, the dosage of estradiol valerate was increased orally, with a maximum of dose of 10 mg/d. When the endometrial thickness was $\geq$ 8 mm, a 100 mg/d progesterone needle was used to transform the endometrium, and resuscitation transplantation was performed 3 days later. (3) The ovulation cycle was deemed suitable for patients with a natural cycle endometrial thickness < 8 mm or menorrhagia. These patients were given 75 U of human menopausal gonadotropin (HMG) by intramuscular injection daily from the fifth day of menstruation and underwent B ultrasound monitoring of follicles and the endometrium after 5 days; the dosage was adjusted as needed. hCG (10,000 U) was given to induce ovulation when the dominant follicle diameter was $\geq$ 18 mm. After ovulation, 40 mg/d progesterone was given to prepare the endometrium for implantation, and 3

days later, the embryo was thawed and transplanted. After transplantation, progesterone (60–80 mg/d) was given.

Serum hCG was detected 14–16 days after transplantation. Serum hCG > 5 U/L was positive. Twenty-eight days after transplantation, guided B ultrasound examination was performed to confirm the presence of an intrauterine pregnancy, and early cardiac motion indicated a clinical pregnancy. Follow-up of the pregnant women and their newborns was performed by telephone.

This study was approved by the Ethics Committee of the Affiliated Hospital of Guangdong Medical University (PJKT2022-093). The conduct of this study is in line with the Declaration of Helsinki.

## Statistical analysis

SPSS 26.0 statistical software was used for data analysis. The adoption rate (%) of enumeration data indicates that the comparison of the rates between groups was performed via the $\chi^2$ test; normally distributed data are expressed as $\bar{X} \pm$ SDs, the independent sample t test was used for comparisons, and the LSD t test was used for pairwise multiple comparisons. Nonnormally distributed data are presented as the median (quartile range) [M (P25–P75)], and the K–W test was used for comparisons. The Mann–Whitney U test was used for comparisons between groups. When the number of events was less than 5, the chi-square test and Fisher's exact test were used to analyze differences between groups, and $P < 0.05$ indicated that the difference was statistically significant.

## Results

### Patient characteristics

Our retrospective study included 566 patients: 104 patients in Group A, 212 in Group B, 102 in Group C and 148 in Group D. The patient characteristics are presented in Table 1. The BMI and bFSH and bLH levels in Groups A and C were higher than those in Groups B and D; the level of $bE_2$ in Group C was higher than that in Group B; The AFC of PCOS patients was higher than that of non-PCOS patients ($P < 0.05$).

### Outcome of ART treatment

There was no significant difference in the total number of Gn days or endometrial thickness on the HCG test day among the groups ($P > 0.05$). However, the $E_2$ level on the HCG test day, the number of retrieved oocytes and the number of high-quality embryos in the ET group were lower than those in the FET group. The number of IVF cycles in Groups B and D was higher than that in Groups A and C, whereas the opposite was true for ICSI cycles. Additionally, the number of cycles in Groups B and D was higher than that in Groups A and C. The number of day 3 embryos transferred in the ET group was significantly higher than that in the FET group, whereas the number of day 5 embryos transferred in the ET group was lower than that in the FET group ($P < 0.05$) (Table 2).

### Clinical pregnancy outcomes

In this study, the clinical pregnancy rate of Group A was significantly greater than those of Groups B and D ($P > 0.05$) (Table 3).

### Perinatal outcomes

The perinatal data revealed 254 cycles of clinical pregnancy and delivery: 56 in Group A, 97 in Group B, 45 in Group C and 56 in Group D. The natural birth rate of Group D was

**Table 1. General comparison of patient characteristics [M (P25, P75), %].**

| Group | ET | | FET | |
|---|---|---|---|---|
| | **ET-PCOS** | **ET-non-PCOS** | **FET-PCOS** | **FET-non-PCOS** |
| | **A (n = 104)** | **B (n = 212)** | **C (n = 102)** | **D (n = 148)** |
| Female age | 29 (26,30) | 30 (28,32) | 29 (27, 32) | 29 (27,32) |
| Infertility duration | 3 (2,5) | 3 (2,5) | 3 (2,5) | 3 (2,5) |
| Primary infertility | 73.07 (76/104) | 58.96 (125/212) | 76.47 (78/102) | 62.84 (93/148) |
| BMI (kg/m$^2$) | 22.10 | 20.52 | 21.78 | 20.57 |
| | (20.47,24.41)$^{\wedge\dagger}$ | (19.25,22.85) | (19.56,24.30)$^{\wedge\dagger}$ | (19.05,22.94) |
| bFSH (IU/L) | 5.97 | 6.54 | 5.79 | 6.25 |
| | (5.05,7.14)$^{\wedge\dagger}$ | (5.62,7.39) | (5,6.82)$^{\wedge\dagger}$ | (5.6,7.38) |
| bE$_2$ (ng/L) | 38.45 | 36.66 | 42.95 | 39.39 |
| | (27.87,56.45) | (25.76,47.53) | (32.78,57.87)$^{\wedge}$ | (29.26,51.47) |
| bLH (IU/L) | 9.57 | 5.19 | 9.37 | 5.68 |
| | (5.92,14.04)$^{\wedge\dagger}$ | (3.91,6.97) | (5.84,14.25)$^{\wedge\dagger}$ | (4.28,7.63) |
| AFC | 25.50 | 18.00 | 26.50 | 21.00 |
| | (24.00,35.75) | (14.00,24.00)$^{*\#\dagger}$ | (24.00,40.00) | (16.00,24.00)$^{*\#}$ |
| Gravidity | | | | |
| 0 | 20.38(76/373) | 33.51(125/373) | 21.18(79/373) | 24.93(93/373) |
| 1 | 17.16(23/134) | 42.54(57/134) | 14.93(20/134) | 25.37(34/134) |
| 2 | 9.76(4/41) | 48.78(20.41) | 4.88(2/41) | 36.59(15/41) |
| 3+ | 5.56(1/18) | 55.56(10/18) | 5.56(1/18) | 33.33(6/18) |
| Parity | | | | |
| 0 | 19.80(98/495) | 36.16(179/495) | 18.79(93/495) | 25.25(125/495) |
| 1 | 8.82(6/68) | 45.59(31/68) | 13.24(9/68) | 32.35(22/68) |
| 2 | 0.00(0/3) | 66.67(2/3) | 0.00(0/3) | 33.33(1/3) |

Compared with Group A

* P < 0.05; compared with Group B

^ P < 0.05; compared with Group C

# P < 0.05; compared with Group D

† P < 0.05. (Indicates the statistical tests used to evaluate the differences between study groups).

significantly lower than that of Group A and Group B, and the cesarean section rate was significantly higher than that of group A and group B (P < 0.05) (Table 4).

## Perinatal complications of pregnant women

A total of 157 pregnant women had clinical deliveries, 234 were followed up, and 11 were lost to follow-up. Among the pregnant women who were followed up, 50 were in group A, 88 were in group B, 42 were in group C, and 54 were in group D. The incidence of placenta previa in group B was significantly lower than that in group D, and the difference was statistically significant (P<0.05). The rates of gestational diabetes mellitus, hypertensive disorders of pregnancy, placenta previa and premature rupture of the membrane were not significantly different among the groups (P>0.05) (Table 5).

## Perinatal outcomes of newborns

There was no significant difference in the incidence of neonatal hospitalization ≥ 3 d, neonatal hypoglycemia or perinatal death among the three groups (P > 0.05). However, in our study, the incidence of fetal distress in Group B was significantly lower than that in Groups C and D,

**Table 2. Comparison of ovulation induction, fertilization and embryo transfer [M (P25, P75), %].**

| Group | ET | | FET | |
|---|---|---|---|---|
| | ET-PCOS | ET-non-PCOS | FET-PCOS | FET-non-PCOS |
| | A (n = 104) | B (n = 212) | C (n = 102) | D (n = 148) |
| Total gonadotropin dose/(IU) | 1625 (1350,1943.75) | 1875 (1500,2334.38) | 1500 (1187.5,1856.25) | 1637.5 (1350,3100) |
| Gn stimulation duration | 11 (9,12) | 11 (10,11.75) | 10 (9,11) | 10 (10,11) |
| $E_2$ at trigger day (pg/ml) | 2272 | 2555 | 4141 | 3269 |
| | $(1742.5,3768.25)^{\#\dagger}$ | $(1813.25,3367.25)^{\#\dagger}$ | (2206,6803.25) | (2418,6286) |
| Endometrial thickness on trigger day (mm) | 12 (11,13) | 12 (11,14) | 12 (10,13) | 12 (11,14) |
| Retrieved oocyte number | $13.5 (10,16)^{\#\dagger}$ | $12 (9,15)^{\#\dagger}$ | 17.5 (13.75,23) | 14 (11,19) |
| Number of good-quality embryos | $3 (1,5)^{\#\dagger}$ | $2 (1,5)^{\#\dagger}$ | 4 (2,7) | 3 (2,6) |
| Fertilization method | | | | |
| IVF | 68.27 (71/104) | $83.96 (178/212)^{*\#}$ | 70.59 (72/102) | $84.46 (125/148)^{*\#}$ |
| ICSI | 31.73 (33/104) | $14.62 (34/212)^{*\#}$ | 29.41 (30/102) | $15.54 (23/148)^{*\#}$ |
| Day of embryo transfer | | | | |
| Day 3 | $79.81 (83/104)^{\#\dagger}$ | $84.43 (179/212)^{\#\dagger}$ | 14.71 (15/102) | 20.27 (30/148) |
| Day 5 | $20.19 (21/104)^{\#\dagger}$ | $15.57 (33/212)^{\#\dagger}$ | 85.29 (87/102) | 79.73 (118/148) |

Compared with Group A

* P < 0.05; compared with Group B

^ P < 0.05; compared with Group C

# P < 0.05; compared with Group D

† P < 0.05. (indicates the statistical tests used to evaluate the differences between study groups).

the incidence of neonatal jaundice in Group D was significantly greater than that in Groups A and B, and the difference was significant (P < 0.05) (Table 6).

## Regression analysis of related factors affecting the clinical pregnancy rate

The results of univariate and multivariate analyses of possible factors affecting the clinical pregnancy rate of infertile patients are presented in Table 7. According to the univariate analysis, bFSH, the number of oocytes retrieved, the number of high-quality embryos, the type of

**Table 3. Comparison of clinical pregnancy outcomes.**

| Group | ET | | FET | |
|---|---|---|---|---|
| | ET-PCOS | ET-non-PCOS | FET-PCOS | FET-non-PCOS |
| | A (n = 104) | B (n = 212) | C (n = 102) | D (n = 148) |
| Clinical pregnancy | $65.38 (68/104)^{\wedge\dagger}$ | 52.83 (112/212) | 59.80 (61/102) | 48.64 (72/148) |
| Live birth rate | 53.85(56/104) | 45.75(97/212) | 44.12(45/102) | 37.84(56/148) |
| Biochemical pregnancy | 8.65 (9/104) | 9.91 (21/212) | 15.69 (16/102) | 10.81 (16/148) |
| Early spontaneous abortion | 13.24 (9/68) | 9.82 (11/112) | 14.75 (9/61) | 15.28 (11/72) |
| Late abortion | 4.41 (3/68) | 3.57 (4/112) | 8.20 (5/61) | 2.78 (2/72) |
| Total abortion rate | 17.65 (12/68) | 13.39 (15/112) | 22.95 (14/61) | 18.06 (13/72) |
| Medium and heavy OHSS | 4.81 (5/104) | 3.30 (7/212) | / | / |

Compared with Group A

* P < 0.05; compared with Group B

^ P < 0.05; compared with Group C

# P < 0.05; compared with Group D

† P < 0.05. (Indicates the statistical tests used to evaluate the differences between study groups).

**Table 4.** Comparison of perinatal data [M (P25, P75), X±SD, %].

| Group | ET | | FET | |
|---|---|---|---|---|
| | ET-PCOS | ET-non-PCOS | FET-PCOS | FET-non-PCOS |
| | A (n = 56) | B (n = 97) | C (n = 45) | D (n = 56) |
| Gestational weeks | 38.5 (38,39) | 39 (38,40) | 39 (38,40) | 39 (38,39.75) |
| Mode of delivery | | | | |
| Eutocia | 64.29 (36/56) | 65.98 (64/97) | 46.67 (21/45) | 39.29 (22/56)*^ |
| Abdominal delivery | 35.71 (20/56) | 34.02 (33/97) | 53.33 (24/45) | 60.72 (34/56)*^ |
| Neonatal sex | | | | |
| Male | 50 (28/56) | 57.73 (56/97) | 46.67(21/45) | 62.5(35/56) |
| Female | 50 (28/56) | 42.27 (41/97) | 53.33(24/45) | 37.5(21/56) |
| Neonatal weight/(kg) | 3.2 (2.81,3.45) | 3.2 (2.9,3.48) | 3.1 (2.9,3.5) | 3.3 (3.0,3.5) |
| Large infant | 3.57 (2/56) | 3.09 (3/97) | 0 (0/45) | 5.35 (3/56) |
| Low birth weight infant | 3.57 (2/56) | 10.71 (6/56) | 8.89 (4/45) | 0 (0/56) |

Compared with Group A

* P < 0.05; compared with Group B

^ P < 0.05; compared with Group C, # P < 0.05; compared with Group D, † P < 0.05. (Indicates the statistical tests used to evaluate the differences between study groups).

embryo transferred and PCOS were significantly correlated with the clinical pregnancy rate. After multivariate analysis, the number of high-quality embryos (OR = 1.119; 95% CI: 1.042–1.201; p = 0.002) was still independent factors associated with the clinical pregnancy rate (Table 7).

## Regression analysis of related factors affecting lowering of the placenta

The possible factors affecting the clinical pregnancy rate of infertile patients were analyzed by univariate and multivariate analyses. In the single-factor analysis, the number of eggs obtained and the method of transplantation were significantly correlated with the low position of the placenta. After multivariate analysis, there was no significant correlation between the included indicators and the incidence of a low placenta (Table 8).

## Regression analysis of related factors affecting fetal distress

The possible factors affecting nonfetal distress were analyzed by univariate and multivariate analyses. According to the univariate analysis, there was a significant correlation between the

**Table 5.** Perinatal complications of pregnant women [%].

| Group | ET | | FET | |
|---|---|---|---|---|
| | ET-PCOS | ET-non-PCOS | FET-PCOS | FET-non-PCOS |
| | A (n = 50) | B (n = 88) | C (n = 42) | D (n = 54) |
| Gestational diabetes mellitus | 18 (9/50) | 13.64 (12/88) | 14.29 (6/42) | 9.26 (5/54) |
| Hypertensive disorder complicating pregnancy | 8 (4/50) | 2.27 (2/88) | 9.52 (4/42) | 5.56 (3/54) |
| Low placental | 2 (1/50) | 1.13 (1/88)† | 7.14 (3/42) | 12.96 (7/54) |
| Placenta previa | 8 (4/50) | 5.68 (5/88) | 7.14 (3/42) | 7.41 (4/54) |
| Premature rupture of the membranes | 4 (2/50) | 6.81 (6/88) | 7.14 (3/42) | 0 (0/54) |

Compared with Group A, * P < 0.05; compared with Group B, ^ P < 0.05; compared with Group C, # P < 0.05; compared with Group D

† P < 0.05. (Indicates the statistical tests used to evaluate the differences between study groups).

**Table 6. Perinatal outcomes of neonates [%].**

| Group | ET | | FET | |
|---|---|---|---|---|
| | ET-PCOS | ET-non-PCOS | FET-PCOS | FET-non-PCOS |
| | A (n = 50) | B (n = 88) | C (n = 42) | D (n = 54) |
| Fetal intrauterine distress | 2 (1/50) | 0 (0/88)[#†] | 9.52 (4/42) | 11.11 (6/54) |
| Neonatal hospitalization $\geq$ 3 days | 16 (8/50) | 19.32 (17/88) | 23.81 (10/42) | 27.78 (15/54) |
| Icterus neonatorum | 12 (6/50) | 13.64 (12/88) | 33.33 (14/42) | 42.59 (23/54)[*^] |
| Neonatal hypoglycemia | 4 (2/50) | 0 (0/88) | 4.76 (2/42) | 1.85 (1/54) |
| Perinatal death | 0 (0/50) | 1.14 (1/88) | 0 (0/42) | 0 (0/54) |

Compared with Group A

* P < 0.05; compared with Group B

^ P < 0.05; compared with Group C

# P < 0.05; compared with Group D

† P < 0.05. (Indicates the statistical tests used to evaluate the differences between study groups).

embryo transfer method and the occurrence of fetal distress. In the univariate analysis, only the embryo transfer method was correlated. To reduce the deviation of the results, the indicators with P < 0.1 in the univariate analysis were included in the multivariate analysis. The embryo transfer method (OR = 0.030; 95% CI: 0.003–0.348; p = 0.005) was still a relevant factor for the incidence of fetal distress (Table 9).

## Regression analysis of related factors affecting neonatal jaundice

The possible factors affecting the occurrence of neonatal jaundice were analyzed by univariate and multivariate analyses. In the univariate analysis, there was a significant correlation between the transplantation method, embryo transfer method and neonatal jaundice. After multivariate analysis, the transplantation method (OR = 0.219; 95% CI: 0.090–0.533; p = 0.001) was still an independent factor affecting the occurrence of neonatal jaundice (Table 10).

**Table 7. Logistic regression analysis of factors affecting the clinical pregnancy rate.**

| Variables | Univariate analysis p value | OR | (95% CI) | Multivariate analysis p value | OR | (95% CI) |
|---|---|---|---|---|---|---|
| Female age | 0.114 | 0.957 | (0.925,1.011) | | | |
| Infertility duration | 0.156 | 0.948 | (0.881,1.021) | | | |
| bFSH | 0.010 | 0.859 | (0.766,0.904) | 0.092 | 0.902 | (0.801,1.017) |
| bLH | 0.325 | 1.019 | (0.982,1.057) | | | |
| bE$_2$ | 0.526 | 0.999 | (0.994,1.003) | | | |
| BMI | 0.065 | 1.057 | (0.997,1.120) | | | |
| AFC | 0.002 | 1.602 | (1.010,1.048) | 0.184 | 1.015 | (0.993,1.037) |
| E$_2$ on trigger day | 0.484 | 1.000 | (1.000,1.000) | | | |
| Endometrial thickness on trigger day | 0.181 | 1.048 | (0.979,1.122) | | | |
| Total gonadotropin dose | 0.821 | 1.000 | (1.000,1.000) | | | |
| Gn stimulation duration | 0.228 | 1.053 | (0.968,1.146) | | | |
| Retrieved oocyte number | 0.005 | 1.043 | (1.013,1.074) | 0.997 | 1.000 | (0.965,1.037) |
| Number of good-quality embryos | 0.000 | 1.134 | (1.067,1.205) | 0.002 | 1.119 | (1.042,1.201) |
| Fertilization method | 0.941 | 0.985 | (0.657,1.477) | | | |
| Delivery method | 0.371 | 1.164 | (0.440,0.885) | | | |
| Day of embryo transfer | 0.020 | 0.671 | (0.480,0.938) | 0.793 | 0.949 | (0.644,1.399) |
| PCOS | 0.008 | 0.624 | (0.051,0.874) | 0.184 | 1.015 | (0.993,1.037) |

**Table 8. Logistic regression analysis of factors affecting low placental position.**

| Variables | Univariate analysis p value | OR | (95% CI) | Multivariate p-value analysis | OR | (95% CI) |
|---|---|---|---|---|---|---|
| Female age | 0.878 | 0.985 | (0.813,1.194) | | | |
| Infertility duration | 0.934 | 0.989 | (0.756,1.293) | | | |
| bFSH | 0.281 | 0.792 | (0.518,1.210) | | | |
| bLH | 0.402 | 0.935 | (0.800,1.094) | | | |
| $bE_2$ | 0.664 | 0.995 | (0.972,1.018) | | | |
| BMI | 0.551 | 1.067 | (0.862,1.321) | | | |
| AFC | 0.705 | 0.988 | (0.927,1.052) | | | |
| $E_2$ on trigger day | 0.722 | 1.000 | (1.000,1.000) | | | |
| Endometrial thickness on trigger day | 0.433 | 0.905 | (0.706,1.160) | | | |
| Total gonadotropin dose | 0.703 | 1.000 | (0.999,1.001) | | | |
| Gn stimulation duration | 0.964 | 0.993 | (0.723,1.363) | | | |
| Retrieved oocyte number | 0.090 | 1.073 | (0.989,1.164) | | | |
| Number of good-quality embryos | 0.663 | 1.041 | (0.869,1.247) | | | |
| Fertilization method | 0.723 | 1.275 | (0.332,4,902) | | | |
| Delivery method | 0.009 | 0.126 | (0.027,0.591) | 0.061 | 0.153 | (0.022,1.089) |
| Day of embryotransfer | 0.048 | 0.211 | (0.045,0.986) | 0.763 | 0.738 | (0.102,5.321) |
| PCOS | 0.664 | 1.313 | (0.384,4.493) | | | |

## Discussion

With the development of assisted reproductive technology, frozen embryo transfer is becoming increasingly common. However, there is no consensus on whether there are differences in pregnancy complications and pregnancy outcomes between frozen embryo transfer and fresh embryo transfer. This retrospective study compared the clinical pregnancy, perinatal and maternal outcomes of women with PCOS after fresh ET cycles and FET cycles. Moreover, to clarify whether PCOS has an impact on perinatal outcomes, we established a control group of

**Table 9. Logistic regression analysis of the influence on fetal distress.**

| Variables | Univariate analysis p value | OR | (95% CI) | Multivariate p-value analysis | OR | (95% CI) |
|---|---|---|---|---|---|---|
| Female age | 0.095 | 0.845 | (0.693,1.030) | 0.125 | 0.850 | (0.691,1.046) |
| Infertility duration | 0.408 | 1.104 | (0.874,1.394) | | | |
| bFSH | 0.684 | 1.090 | (0.721,1.648) | | | |
| bLH | 0.494 | 1.040 | (0.930,1.162) | | | |
| $bE_2$ | 0.685 | 0.995 | (0.971,1.019) | | | |
| BMI | 0.736 | 1.040 | (0.830,1.303) | | | |
| AFC | 0.887 | 1.005 | (0.944,1.069) | | | |
| $E_2$ on trigger day | 0.056 | 1.000 | (1.000,1.000) | 0.416 | 1.000 | (1.000,1.000) |
| Endometrial thickness on trigger day | 0.820 | 0.971 | (0.752,1.253) | | | |
| Total gonadotropin dose | 0.852 | 1.000 | (0.999,1.001) | | | |
| Gn stimulation duration | 0.118 | 1.235 | (0.948,1.609) | | | |
| Retrieved oocyte number | 0.826 | 0.989 | (0.895,1.093) | | | |
| Number of good-quality embryos | 0.338 | 1.092 | (0.912,1.309) | | | |
| Fertilization method | 0.342 | 0.365 | (0.046,2.919) | | | |
| Delivery method | 0.009 | 0.063 | (0.008,0.499) | 0.005 | 0.030 | (0.003,0.348) |
| Day of embryo transfer | 0.192 | 0.407 | (0.105,1.572) | 0.127 | 3.670 | (0.660,20.397) |
| PCOS | 0.670 | 0.768 | (0.227,2.592) | | | |

**Table 10. Logistic regression analysis of the incidence of neonatal jaundice.**

| Variables | Univariate analysis p value | OR | (95% CI) | Multivariate analysis p value | OR | (95% CI) |
|---|---|---|---|---|---|---|
| Female age | 0.430 | 1.042 | (0.941,1.153) | | | |
| Infertility duration | 0.341 | 1.066 | (0.935,1.214) | | | |
| bFSH | 0.675 | 1.046 | (0.847,1.292) | | | |
| bLH | 0.904 | 1.004 | (0.942,1.070) | | | |
| $bE_2$ | 0.807 | 0.999 | (0.989,1.008) | | | |
| BMI | 0.071 | 0.893 | (0.789,1.010) | | | |
| AFC | 0.951 | 0.999 | (0.967,1.031) | | | |
| $E_2$ on trigger day | 0.601 | 1.000 | (1.000,1.000) | | | |
| Endometrial thickness on trigger day | 0.936 | 1.005 | (0.886,1.141) | | | |
| Total gonadotropin dose | 0.894 | 1.000 | (1.000,1.000) | | | |
| Gn stimulation duration | 0.773 | 1.024 | (0.871,1.203) | | | |
| Retrieved oocyte number | 0.639 | 1.012 | (0.964,1.061) | | | |
| Number of good-quality embryos | 0.311 | 1.050 | (0.955,1.155) | | | |
| Fertilization method | 0.342 | 0.680 | (0.307,1.507) | | | |
| Delivery method | 0.000 | 0.239 | (0.126,0.455) | 0.001 | 0.219 | (0.090,0.533) |
| Day of embryotransfer | 0.007 | 0.414 | (0.218,0.787) | 0.779 | 1.139 | (0.459,2.827) |
| PCOS | 0.608 | 1.178 | (0.630,2.201) | | | |

women with tubal infertility. A comparison of the differences in pregnancy outcomes and pregnancy complications between fresh embryo transfer and frozen embryo transfer revealed that PCOS patients could achieve a higher clinical pregnancy rate for a singleton fetus by fresh embryo transfer; moreover, frozen-thawed embryo transfer may increase the incidence of neonatal jaundice and fetal distress in singleton pregnancies. Multivariate analysis revealed that the number of high-quality embryos and the incidence of PCOS were independent factors affecting clinical pregnancy, and the embryo transfer method was an independent factor affecting fetal distress and neonatal jaundice.

In recent years, complications related to assisted reproductive technology have attracted increasing attention. According to the World Report of the International Commission on Monitoring Assisted Reproductive Technology [9–12], the pregnancy loss rates in fresh embryo transfer cycles were 21.8%, 21.1% and 20.2% in 2008, 2009 and 2010, respectively. The early pregnancy loss rates corresponding to FET were 28.9% (2008), 25.4% (2009) and 25.2% (2010). In a study by Carlos Simón et al. [13], the abortion rate after fresh embryo transfer was 5.4%, and that after FET was 14%. A study by Carlos Simón et al. reported that the abortion rate after fresh embryo transfer was 5.4%, whereas that after FET was 14%. The pregnancy loss rate of frozen-thawed embryo transfer was significantly higher than that of fresh embryo transfer. This finding was also verified in our study. Here, we also found that the incidence of early abortion and total abortion in patients who underwent frozen-thawed embryo transfer was higher than that in patients who underwent fresh embryo transfer. The potential cause of early abortion may be the damage related the freezing of embryos during the freezing process. The freezing and thawing process may cause instability of the embryonic genes, an increase in embryonic DNA fragments, and an increase in mitochondrial mutations, resulting in a decrease in the developmental potential of embryos. [14] At the same time, epigenetic changes may also occur in the embryo during the freezing and thawing process, which will have an adverse effect on embryonic development, resulting in an increase in the abortion rate. [14]. Moreover, we found that the clinical pregnancy rate of fresh embryo transfer is higher than that of frozen-thawed embryo transfer, which biases the results in favor of fresh embryo

transfer. In contrast to most studies, other studies have reported that the clinical pregnancy rate of frozen-thawed embryo transfer is higher than that of fresh embryo transfer. They showed that in fresh ET cycles, the use of controlled superovulation technology makes the determination of the superphysiological level of estradiol in IVF/ICSI cycles inevitable. High concentrations of $E_2$ can affect embryo implantation, placental development and the uterine blood supply [15, 16]. Other studies have shown that when $E_2$ is > 3000 pg/ml, the risk is significantly increased [17]. However, in our study, the level of $E_2$ on the trigger day of fresh embryo transfer patients was mostly lower than 3000 pg/ml, and the level of $E_2$ on the trigger day of PCOS patients was lower than that of non-PCOS patients. Moreover, we also found that the clinical pregnancy rate of PCOS patients was higher than that of tubal factor infertility patients (Group A > Group B > Group D, P < 0.05), which may be related to the fact that PCOS patients have more basal follicles, can produce more eggs and can cultivate more high-quality embryos when treated with controlled ovarian hyperstimulation. We also found that the number of high-quality embryos (OR = 1.119; 95% CI: 1.042–1.1201; p = 0.002) was independent factors affecting clinical pregnancy.

Increasing evidence supports the association between IVF and the risk of adverse perinatal outcomes, with analysis suggesting that FET is not superior to fresh ET and that the placenta has long been considered a key predictor of maternal and offspring health [18]. Guo et al. [19] included 1516 women who gave birth to a singleton fetus. The results showed that compared with fresh embryo transfer, frozen-thawed embryo transfer was associated with an increased risk of placenta previa and placental adhesion, indicating that frozen-thawed embryo transfer was an independent risk factor for placental adhesion. Compared with other methods, frozen-thawed embryo transfer is more likely to be associated with blastocyst transfer. When IVF treatment is implemented, poor blastocyst rotation may lead to low placental position and placenta previa. The findings of our study are consistent with those of Guo et al. In our study, the probability of a low placenta in patients who underwent frozen-thawed embryo transfer was higher than that in patients who underwent the fresh embryo transfer(Group D > Group B, P < 0.05). However, after multivariate correction analysis, the embryo transfer method was not an independent factor influencing the occurrence of a low placental position (OR = 0.153). 95% CI: 0.022–1.089; p = 0.061).

In addition, in our study, we found that women who underwent frozen-thawed embryo transfer were more likely to deliver by cesarean section than were women who underwent fresh embryo transfer. This may be related to the psychological course and perinatal examination of patients undergoing treatment by assisted reproductive technology for frozen-thawed embryo transfer. FET greatly prolonged the treatment time, thus prolonging the live birth time, which not only required additional embryo operations but also increased the cost and workload of treatment and increased the degree of psychological pressure on patients. This psychological pressure makes them think that cesarean section is a safer method of delivery, and obstetricians may think that newborns conceived by assisted reproductive technology are more 'precious'. Multiple comprehensive factors lead to a higher cesarean section rate.

With the development of assisted reproductive technology, freezing and thawing technology has been effectively improved, but freezing injury in embryos is still inevitable. Whether embryo transfer has adverse effects on offspring is controversial. With respect to neonatal perinatal outcomes, studies have shown that FET and fresh ET have no significant effect on neonatal perinatal outcomes. However, in our study, we investigated the incidence of fetal distress (Group B < Group C, Group D, P < 0.05) and neonatal jaundice (Group D > Group A, Group B, P < 0.05) after frozen-thawed embryo transfer. Multivariate analysis revealed that embryo transfer was an independent factor affecting the incidence of neonatal jaundice (OR = 0.219; 95% CI: 0.090–0.533; p = 0.001). This finding is basically consistent with the

findings of Jiarong Li et al. [20]. However, these diseases have not been reported to be related to FET, and whether they have a potential impact on embryos during the freezing process is still inconclusive. Moreover, owing to the advantages and disadvantages of FET and fresh embryo transfer, further observations and explorations are needed.

In addition, in our study, we found that women who received frozen-thawed embryo transfer were more likely to deliver by cesarean section than women who received fresh embryo transfer, which may be related to the multiple transfers of embryos to pregnant women who received frozen-thawed embryo transfer. This psychological pressure makes them believe that cesarean section is a safer method of delivery; therefore, they prefer cesarean section for delivery.

## Conclusions

In summary, young PCOS patients without risk of OHSS have a high clinical pregnancy rate with fresh transplant cycles. PCOS disease itself has no significant effect on the perinatal outcomes of the mother or singleton infant. Frozen-thawed embryo transfer may increase the incidence of low placenta, fetal distress and neonatal jaundice. Further attention needs to be given to the impact of embryo transfer methods on perinatal outcomes to optimize assisted reproductive technology, strengthen pregnancy management and screening, and ensure the health of mothers and children after assisted pregnancy. The advantages and disadvantages of fresh or frozen-thawed embryo transfer still need long-term exploration and research.

## Supporting information

**S1 Data. Datasets used in the research.**
(ZIP)

## Author Contributions

**Conceptualization:** Tianzhong Ma.

**Formal analysis:** Yanru Niu.

**Methodology:** Xia Zhu, Xiaomei Gao.

**Resources:** Lihua Xu.

**Writing – original draft:** Huizhen Li, Tianzhong Ma.

**Writing – review & editing:** Tianzhong Ma.

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
