## [Decision Letter · Decision Letter 0]

7 Jun 2024

PONE-D-24-09843The effect of fresh embryo transfer and frozen-thawed embryo transfer on the perinatal outcomes of single fetuses from mothers with PCOSPLOS ONE

Dear Dr. ma,

Thank you for submitting your manuscript to PLOS ONE. After careful consideration, we feel that it has merit but does not fully meet PLOS ONE’s publication criteria as it currently stands. Therefore, we invite you to submit a revised version of the manuscript that addresses the points raised during the review process.

We look forward to receiving your revised manuscript.

Kind regards,

Tamara Sljivancanin Jakovljevic

Academic Editor

PLOS ONE

Journal Requirements:

3. Thank you for stating the following financial disclosure: "This experiment was supported by funding from the National Natural Science Foundation of China (81300484) and the Natural Science Foundation of Guangdong Province, China (2022A1515010849). "

Reviewers' comments:

Reviewer's Responses to Questions

**Comments to the Author**

1. Is the manuscript technically sound, and do the data support the conclusions?

Reviewer #1: Partly

Reviewer #2: No

2. Has the statistical analysis been performed appropriately and rigorously? 

Reviewer #1: Yes

Reviewer #2: No

3. Have the authors made all data underlying the findings in their manuscript fully available?

Reviewer #1: Yes

Reviewer #2: Yes

4. Is the manuscript presented in an intelligible fashion and written in standard English?

Reviewer #1: No

Reviewer #2: Yes

5. Review Comments to the Author

Reviewer #1: Dear authors, thank you for submitting your manuscript for consideration for publication in PLOS ONE. I find some issues that have to be address in order to improve the quality of the manuscript. You have written "The inclusion criteria for the PCOS group were as follows: less than 35 years of age; retrieval of more than 5 eggs; selection of a long protocol". Could you please explain (justify) the selection of those inclusion criteria, having in mind:

1) "retrieval of more than 5 eggs" in PCOS is considered as poor ovarian response in PCOS and it is happening commonly, especially in women with severe PCOS (very high AMH)

2) "selection of a long protocol" is usually not considered for women with PCOS

3) "less than 35 years of age" – significant proportion of women with PCOS undergoing IVF belongs to the population older than 35 years of age

Please rewrite the sentence “According to the transplantation method”, since the "transplantation method" could be replaced with more suitable word. The same is for the “the embryo was thawed and transplanted 3 days later" and it could be better written as “the embryo was thawed and transferred 3 days later". Furthermore, I believe you should find better word / expression for "intima thickness".

I cannot understand in row one in Table 1 "Female sex". What kind of f patient characteristic of women undergoing IVF is "Female sex". Furthermore, in order to simplify the Table 1, you could leave out row with "Secondary infertility". Moreover, in footnote of the table you could indicate the statistical tests used to evaluate the differences between study groups. The same should be done for Table 2, 3 and 4 (indicate the statistical tests used to evaluate the differences between study groups).

In Table 2, please "Total Gn" replace with "Total gonadotropin dose", "Gn Total days" with "stimulation duration", "HCG testing day E2/(pg/ml)" with "E2 at trigger day", "Endometrial thickness on HCG testing day/(mm)" with "Endometrial thickness on trigger day/(mm)", "Proportion of embryo transfers" with "Day of embryotransfer".

How do you explain impossible results in your study presented in Table 3: Clinical pregnancy rates being higher than Biochemical pregnancy rates (68 pregnant out of 104 in Group A, 112 pregnant out of 212 in Group B…. Biochemical pregnancy (9 pregnant out 104 in Group A, 21 pregnant out of 212 in Group B, etc)…

Discussion is poorly written. The beginning of Discussion: "The use of assisted reproductive technology has increased steadily in the past 30 years, helping many infertile families. However, the abnormal internal environment during ART treatment may increase the risk of health problems in future generations. Among them, higher maternal E2 levels are notable. Due to the use of COH technology, an increased E2 level in supraphysiological growth hormone during IVF/ICSI cycles is inevitable. However, with the advancement of embryo cryopreservation technology, especially in vitrification, FET enables women receiving COH to maintain their initial hormone levels before embryo transfer, but whether this can improve adverse pregnancy and perinatal outcomes remains unclear" belong more to Introduction section or to some book chapter, not to Introduction. At the beginning instead of your first sentences summarize key results with reference to study objectives. Then, give a cautious overall interpretation of results considering objectives, limitations, multiplicity of analyses, results from similar studies, and other relevant evidence.

Reviewer #2: Numerous changes needed to be made:

1.Table 1 needs more data such as patient age, gravity, parity, history of prior c-section - these are ALL factors that are crucial to understanding perinatal risk. Without this data, the conclusions of the study remain faulty.

2. The above factors need to be assessed. if significant differences are seen, then they need to be included in a multivariable logistic regression to assess their influence on embryo transfer type on obstetrical outcomes.

Without the above, the paper should not be published.

6. PLOS authors have the option to publish the peer review history of their article (what does this mean?). If published, this will include your full peer review and any attached files.

Reviewer #1: No

Reviewer #2: No

---

## [Author Response · Author response to Decision Letter 0]

12 Jul 2024

Dear Editors and Reviewers：

Thank you for your letter and for the reviewers' comments concerning our manuscript entitled “The effect of fresh embryo transfer and frozen-thawed embryo transfer on the perinatal outcomes of single fetuses from mothers with PCOS” (ID: PONE-D-24-09843）. Those comments are all valuable and very helpful for revising and improving our paper, as well as the important guiding significance to our researches. We have studied comments carefully and have made correction which we hope meet with approval. Revised portion are marked in yellow in the paper. The main corrections in the paper and the responds to the reviewer’s comments are as flowing：

Responds to the reviewer's comments:

Reviewer #1: 

1)"retrieval of more than 5 eggs" in PCOS is considered as poor ovarian response in PCOS and it is happening commonly, especially in women with severe PCOS (very high AMH).

Response："retrieval of more than 5 eggs" is the inclusion criteria for this study, in order to exclude patients with severe poor ovarian function. If patients with PCOS had less than 5 eggs extracted after superovulation, these patients were not included in our study.

2)"selection of a long protocol" is usually not considered for women with PCOS.

Response：This study is a retrospective study. The initial development of the center is mainly based on the use of long-term regimen, and it is gradually converted to antagonist regimen after the treatment technology is mature. Therefore, the long scheme is selected as the research entry point. 

3)"less than 35 years of age" – significant proportion of women with PCOS undergoing IVF belongs to the population older than 35 years of age.

Response：China 's national conditions define age ≥ 35 years old as elderly patients. In order to avoid the influence of elderly factors, this study screened patients younger than 35 years old.

4)Please rewrite the sentence“According to the transplantation method”, since the "transplantation method" could be replaced with more suitable word. The same is for the “the embryo was thawed and transplanted 3 days later" and it could be better written as “the embryo was thawed and transferred 3 days later". Furthermore, I believe you should find better word / expression for "intima thickness".

Response：We modified ' according to the method of transplantation ' to ' according to whether the fresh embryo transfer patients were divided into fresh embryo transfer group and frozen-thawed embryo transfer group ' ; revise ' the embryo was thawed and transplanted 3 days later ' to ' the embryo was thawed and transferred 3 days later' ; the ' intima thickness ' was modified to ' endometrial thickness '.

5)I cannot understand in row one in Table 1 "Female sex". What kind of f patient characteristic of women undergoing IVF is "Female sex". Furthermore, in order to simplify the Table 1, you could leave out row with "Secondary infertility". Moreover, in footnote of the table you could indicate the statistical tests used to evaluate the differences between study groups. The same should be done for Table 2, 3 and 4 (indicate the statistical tests used to evaluate the differences between study groups).

Response：I am very sorry for such a low-level error.I changed ' Female sex ' to ' Female age '.Secondary infertility data were omitted in accordance with expert opinion and statistical tests used to to evaluate the differences between study groups were indicated under the table.

6)In Table 2, please "Total Gn" replace with "Total gonadotropin dose", "Gn Total days" with "stimulation duration", "HCG testing day E2/(pg/ml)" with "E2 at trigger day", "Endometrial thickness on HCG testing day/(mm)" with "Endometrial thickness on trigger day/(mm)", "Proportion of embryo transfers" with "Day of embryotransfer".

Response：According to the reviewer 's opinion, we modified ' Total Gn ' to ' Total gonadotropin dose ' ; the ' Gn Total days ' was modified to ' Gn stimulation duration ', and the ' HCG testing day E2 / ( pg / ml ) ' was modified to ' E2 at trigger day '. The ' Endometrial thickness on HCG testing day / ( mm ) ' was modified to ' Endometrial thickness on trigger day / ( mm ) ', and the ' Proportion of embryo transfers ' was modified to ' Day embryo of embryo transfer '.

7)How do you explain impossible results in your study presented in Table 3: Clinical pregnancy rates being higher than Biochemical pregnancy rates (68 pregnant out of 104 in Group A, 112 pregnant out of 212 in Group B…. Biochemical pregnancy (9 pregnant out 104 in Group A, 21 pregnant out of 212 in Group B, etc)

Response：The Biochemical pregnancy in this study is defined as the number of pregnancies diagnosed only by beta-HCG detection without a gestational sac visualized by vaginal ultrasound at the fifth week of pregnancy, per number of pregnancies.

8)Discussion is poorly written. The beginning of Discussion: "The use of assisted reproductive technology has increased steadily in the past 30 years, helping many infertile families. However, the abnormal internal environment during ART treatment may increase the risk of health problems in future generations. Among them, higher maternal E2 levels are notable. Due to the use of COH technology, an increased E2 level in supraphysiological growth hormone during IVF/ICSI cycles is inevitable. However, with the advancement of embryo cryopreservation technology, especially in vitrification, FET enables women receiving COH to maintain their initial hormone levels before embryo transfer, but whether this can improve adverse pregnancy and perinatal outcomes remains unclear" belong more to Introductio

Response：We are very sorry for our negligence in the writing of the discussion section. We rewrite the discussion section according to the suggestions of the reviewers, and thank the reviewers for their hard work again.

Reviewer #2： 

1)Table 1 needs more data such as patient age, gravity, parity, history of prior c-section - these are ALL factors that are crucial to understanding perinatal risk. Without this data, the conclusions of the study remain faulty.

Response：In this study, patients with a history of uterine surgery were excluded before data analysis. The age of female infertility was calculated in Table 1, and BMI was calculated in Table 2.

2)The above factors need to be assessed. if significant differences are seen, then they need to be included in a multivariable logistic regression to assess their influence on embryo transfer type on obstetrical outcomes

Response：Based on the opinions of the reviewers, single factor and multi-factor analysis were carried out. The content has been added to the text to thank the experts for their suggestions.

We tried our best to improve the manuscript and made some changes in the manuscript.These changes will not influence the content and framework of the paper.And here we did not list the changes but marked in yellow in revised paper.

We appreciate for Editors/Reviewers’ warm work earnestly, and hope that the correction will meet with approval.

Once again， thank you very much for your comments and suggestions.

Sincerely,

Tianzhong Ma and Huizhen Li

---

## [Decision Letter · Decision Letter 1]

6 Aug 2024

PONE-D-24-09843R1The effect of fresh embryo transfer and frozen-thawed embryo transfer on the perinatal outcomes of single fetuses from mothers with PCOSPLOS ONE

Dear Dr. ma,

Thank you for submitting your manuscript to PLOS ONE. After careful consideration, we feel that it has merit but does not fully meet PLOS ONE’s publication criteria as it currently stands. Therefore, we invite you to submit a revised version of the manuscript that addresses the points raised during the review process.

We look forward to receiving your revised manuscript.

Kind regards,

Tamara Sljivancanin Jakovljevic

Academic Editor

PLOS ONE

Reviewers' comments:

Reviewer's Responses to Questions

**Comments to the Author**

1. If the authors have adequately addressed your comments raised in a previous round of review and you feel that this manuscript is now acceptable for publication, you may indicate that here to bypass the “Comments to the Author” section, enter your conflict of interest statement in the “Confidential to Editor” section, and submit your "Accept" recommendation.

Reviewer #1: All comments have been addressed

Reviewer #2: (No Response)

2. Is the manuscript technically sound, and do the data support the conclusions?

Reviewer #1: Yes

Reviewer #2: Partly

3. Has the statistical analysis been performed appropriately and rigorously? 

Reviewer #1: Yes

Reviewer #2: Yes

4. Have the authors made all data underlying the findings in their manuscript fully available?

Reviewer #1: (No Response)

Reviewer #2: Yes

5. Is the manuscript presented in an intelligible fashion and written in standard English?

Reviewer #1: (No Response)

Reviewer #2: Yes

6. Review Comments to the Author

Reviewer #1: (No Response)

Reviewer #2: Gravity and Parity are still missing - these are certainly factors that could influence the conclusions of this paper. It is concerning that the authors did not incorporate this despite requesting this after review of their original manuscript.

Also, antral follicle counts and/or AMH levels should also be compared between all of the groups.

7. PLOS authors have the option to publish the peer review history of their article (what does this mean?). If published, this will include your full peer review and any attached files.

Reviewer #1: No

Reviewer #2: No

---

## [Author Response · Author response to Decision Letter 1]

26 Aug 2024

Dear Reviewers：

Thank you for your letter and for the reviewers' comments concerning our manuscript entitled “The effect of fresh embryo transfer and frozen-thawed embryo transfer on the perinatal outcomes of single fetuses from mothers with PCOS” (ID: PONE-D-24-09843R2）. Those comments are all valuable and very helpful for revising and improving our paper, as well as the important guiding significance to our researches. We have studied comments carefully and have made correction which we hope meet with approval. Revised portion are marked in yellow in the paper. The main corrections in the paper and the responds to the reviewer’s comments are as flowing：

Responds to the reviewer's comments:

Reviewer#2: 

1.Gravity and Parity are still missing - these are certainly factors that could influence the conclusions of this paper. It is concerning that the authors did not incorporate this despite requesting this after review of their original manuscript.

Response：Thank you for the insightful and constructive feedback provided by the reviewers. In response, we have enriched our manuscript with pertinent supplementary data（AFC, Gravity and Parity）, which are distinctly highlighted in yellow for ease of reference within the revised article.

We appreciate for Editors/Reviewers’ warm work earnestly, and hope that the correction will meet with approval.

Once again， thank you very much for your comments and suggestions.

Sincerely,

Tianzhong Ma and Huizhen Li

---

## [Decision Letter · Decision Letter 2]

30 Sep 2024

The effect of fresh embryo transfer and frozen-thawed embryo transfer on the perinatal outcomes of single fetuses from mothers with PCOS

PONE-D-24-09843R2

Dear Dr. Tianzhong Ma,

We’re pleased to inform you that your manuscript has been judged scientifically suitable for publication and will be formally accepted for publication once it meets all outstanding technical requirements.

Kind regards,

Tamara Sljivancanin Jakovljevic

Academic Editor

PLOS ONE

Additional Editor Comments (optional):

Reviewers' comments:

Reviewer's Responses to Questions

**Comments to the Author**

1. If the authors have adequately addressed your comments raised in a previous round of review and you feel that this manuscript is now acceptable for publication, you may indicate that here to bypass the “Comments to the Author” section, enter your conflict of interest statement in the “Confidential to Editor” section, and submit your "Accept" recommendation.

Reviewer #2: All comments have been addressed

2. Is the manuscript technically sound, and do the data support the conclusions?

Reviewer #2: Yes

3. Has the statistical analysis been performed appropriately and rigorously? 

Reviewer #2: Yes

4. Have the authors made all data underlying the findings in their manuscript fully available?

Reviewer #2: Yes

5. Is the manuscript presented in an intelligible fashion and written in standard English?

Reviewer #2: Yes

6. Review Comments to the Author

Reviewer #2: Thank you for making the previously-mentioned correction. The incorporation of gravity and parity greatly increase the robustness of your conclusions relating to perinatal outcomes.

---

## [Editor Report · Acceptance letter]

16 Oct 2024

PONE-D-24-09843R2 

PLOS ONE

Dear Dr. ma, 

I'm pleased to inform you that your manuscript has been deemed suitable for publication in PLOS ONE. Congratulations! Your manuscript is now being handed over to our production team.

Kind regards, 

on behalf of

Dr. Tamara Sljivancanin Jakovljevic 

Academic Editor

PLOS ONE